# Stress and job satisfaction over time, the influence of the managerial position: A bivariate longitudinal modelling of Wittyfit data

Rémi Colin-Chevalier[1]*, Frédéric Dutheil[1], Amanda Clare Benson[2], Samuel Dewavrin[3], Thomas Cornet[3], Céline Lambert[4], Bruno Pereira[4]

1 Université Clermont Auvergne, CNRS, LaPSCo, CHU Clermont-Ferrand, Cegid, Clermont-Ferrand, France, 2 Sport Innovation Research Group, Department of Health and Biostatistics, Swinburne University of Technology, Hawthorn, Victoria, Australia, 3 Cegid, Lyon, France, 4 Biostatistics Unit, DRCI, CHU Clermont-Ferrand, Clermont-Ferrand, France

* r.colin6374@gmail.com

## Abstract

### Background

The managerial position affects stress and job satisfaction of workers, but these influences have always been studied separately.

### Objective

We aimed to assess bivariate influence of the managerial position on workers' stress and job satisfaction and the inter-relationship of these indicators over time.

### Methods

We have analyzed data from workers who use the Wittyfit software, collected annually between 2018 and 2021. Stress and job satisfaction were evaluated by self-report questionnaires. Job position (manager or employee) was provided by the software's client companies.

### Results

Data of 704 workers were included in the study. Cross-sectional and longitudinal multivariate analyses revealed that managerial position improves job satisfaction (p<0.001), but not stress (p = 0.4). Overall, while workers' job satisfaction has improved (p<0.001), stress has remained stable over time (p = 0.3). Three latent groups, with specific evolutionary multi-trajectory of stress and job satisfaction were identified in the sample (entropy = 0.80). Age and seniority, but not gender tended to influence managers' and employees' indicators. Over time, stress and job satisfaction have tended to negatively interconnect, in cross-section and in a cross-lagged manner (p<0.001).

**Data Availability Statement:** All relevant data are within the paper and its Supporting Information files.

**Funding:** The authors received no specific funding for this work.

**Competing interests:** RC, SD, and TC are part of Wittyfit. Other authors have declared that no competing interests exist (FD is responsible for the scientific accuracy of Wittyfit but is not paid by Cegid; as previously published, Wittyfit is a public private partnership with the CHU of Clermont-Ferrand, France). This does not alter our adherence to PLOS ONE policies on sharing data and materials.

## Conclusions

The managerial position improves workers' job satisfaction but has no effect on stress. Sociodemographics including age and seniority, but not gender, can affect this relationship. Stress and job satisfaction can influence each other, both cross-sectionally and over time. To be more effective, organizations should implement holistic strategies targeting multiple indicators.

## Trial registration

Clinicaltrials.gov: NCT02596737.

## Introduction

Stress refers to the inability of an organism to face a threat, whether physical or emotional [1], and its physiological and pathophysiological response [2]. Known to increase the risk of cardiovascular disease [3,4], multiple factors, mostly work-related, have an influence on stress [5,6]. Stress has become a major problem for organizations [7]. Job satisfaction is a state of balance between a worker's expectations for his or her job and the professional values that the job enables him or her to achieve [8]. As an emerging and predictive factor of good occupational health [9], several studies have highlighted its main work-related drivers [10,11]. In particular, stress and job satisfaction were found to have a negative influence on each other, in other words, stress is a factor of job dissatisfaction [12–14].

The managerial position was found to be a common factor in stress [15] and job satisfaction [16], as were sociodemographic factors such as age, seniority and gender [16–18]. To the best of our knowledge, no evaluation of the putative short- and long-term influences of managerial status on occupational health indicators such as stress and job satisfaction have been conducted yet. The managerial role being associated with greater duties and responsibilities [19], but also more risk for health [20,21], it seems relevant to understand whether managers may be affected differently from employees, how this persists over time, and how age, seniority or gender may affect workers' stress and job satisfaction depending on their job position.

Thus, the main objective of our study was to evaluate the influence of the job position, notably the managerial role, on stress and job satisfaction. The secondary objectives were to explore the simultaneous evolution over time of the two work-related indicators and to assess their reciprocal influences, in all workers and as a function of the job position (manager or employee), and to measure the influence of age, seniority and gender on workers' stress and job satisfaction according to their job position.

## Materials and methods

### Recruitment

Wittyfit is a digital web software designed by Cegid (Lyon, France) in partnership with the university hospital of Clermont-Ferrand, implemented in numerous French companies from any industry sector [22]. Wittyfit enables the well-being of a company's worker to be assessed by collecting data on their feelings to improve the collective performance. Based on voluntary participation, the software offers its users the opportunity to express themselves anonymously by answering various health-related questionnaires to evaluate their overall health status and provides personalized feedback. This study received approval from both the National

Commission for Data Protection and Liberties (CNIL) and the ethics committee of South-East VI (identified on clinicaltrials.gov with the registration number NCT02596737).

## Material

**Stress and job satisfaction.** During the data collection period, users of the Wittyfit software could assess for the first time or update their stress and job satisfaction levels using the relative analog scales, as often as they wished [23–25]. The software's visual analog scales go from 0 (low level) to 100 (high level). For both parameters, the average annual score of a worker was computed and retained so that each worker was defined by only two records per year, one for stress and one for job satisfaction. If a worker did not complete a questionnaire for a full year, it was considered a missing value.

**Time.** Time was defined as the year of the measurements, ranging from 2018 to 2021, taken in their entirety. The first year of the study, 2018, was considered as the reference point for comparing changes in stress and job satisfaction in the longitudinal analyses.

**Job position and other sociodemographics.** Wittyfit collects sociodemographic data from its users. This includes information about their job position, which can only have two states: manager or employee. Managers can be defined as workers in charge of an organization, with one or more individuals under their supervision, namely the employees. In addition to job position, Wittyfit also collects the age, seniority, gender (male or female) and company of its users. Age has been transformed into a two-modality class depending on whether the worker was above or below the age of 40-years old. Similarly, seniority has been transformed into a two-modality class depending on whether the worker was in a position of seniority for greater or less than five years. These thresholds were chosen in accordance with the median of these indicators in the sample.

**Inclusion/exclusion criteria.** Workers from multiple French companies using Wittyfit were included in the study. Data collection began in 2018 and continued through 2022 (excluded). All companies that were not yet Wittyfit customers in 2018 have been excluded, as well as those with no managers, as it was impossible to measure the effect of job position in these companies. To be included, a worker had to have at least one record between stress and job satisfaction in time one. Workers with two or fewer annual records of stress or job satisfaction were excluded because it is impossible to estimate individual change with too few measures [26]. Users with no sociodemographic data were excluded.

## Statistical analysis

Statistical analyses were performed using R (version 4.2.1) and Stata (version 11.2) software. Quantitative data (stress and job satisfaction per year), expressed as mean±SD, were compared between managers and employees (i) in cross-section using first a univariate and then a bivariate linear mixed model with only company effect as random and (ii) longitudinally, still bivariately, adding individual and time random effects. Subgroup analyses were conducted to assess the influence of other sociodemographic data on both outcomes by job position. Then, longitudinal analysis was conducted using multivariate generalized linear mixed models to evaluate the workers' average changes in stress and job satisfaction over time. Afterwards, the same analysis was conducted for each job position, and stress and job satisfaction were compared at each time. A group-based multi-trajectory model [27], using the "traj, multgroups" function of the "traj" package of Stata, was used to highlight latent groups within the sample, according to their changes in stress and job satisfaction. Model selection has been made by following multiple rules: (1) each trajectory had to have an average posterior probability of assignment greater than 70%, (2) a probability of correct classification greater than five, (3) include more than 5%

of individuals in the total sample, and (4) have an entropy greater than or equal to 0.80 [28,29]. The model was fitted for several classes ranging from one to 10, and the one with the largest number of groups and which complied with the rules was selected. Individuals were then assigned to the group to which they were most likely to belong, that is the group whose trajectory was most like their own individual trajectory. A multinomial logistic mixed model with company effect as random enabled comparison of the proportion of managers and employees into the different groups. Finally, we used a cross-lagged panel model to assess the simultaneous relationships between stress and job satisfaction over time, among all workers and then by job position.

All models' estimates were turned into Hedges' g (also called g) effect sizes [30] and 95% confidence intervals (95CI) and interpreted according to Cohen's rules [31]. Multicollinearity of outcomes was checked prior to analysis. Linear model assumptions (residual independence and normality, and variance homogeneity) were verified post hoc. Missing data phenomenon has occurred over the course of the study. As the type of missing data was identified as not missing at random, and the structure of missing data as non-monotonic, a sensitivity analysis was conducted. Five different datasets were used to assess the sensitivity of the results: (i) an available-cases dataset, (ii) a complete-cases dataset (i.e., the dataset that includes workers who reported their stress and job satisfaction levels each year), two single-imputed datasets imputed with (iii) the linear interpolation and (iv) the "last observation carried forward" methods, and (v) a multiple-imputed dataset. Single imputations were performed with the command "imputation" of the "longitudinalData" package of R. Multiple imputation were performed with the "mice" command of the eponymous package of R. Only estimates (g and 95CI) obtained with the original dataset (the dataset of available cases) were included. Unless specified, a p-value <0.05 was considered as statistically significant.

Further information on the conduct of the study and the quality of reporting was provided in the distribution of outcomes (S1 Appendix), the statistical analysis plan (S2 Appendix), the sensitivity analysis results (S3 Appendix), and the reporting guidelines (S4 Appendix).

## Results

### Participants

Data were collected from 704 workers (362 men, 342 women), with 381 workers over 40, and 434 with more than 5 years' seniority at the start of the study. Workers included 81 managers and 623 employees from three different companies, with 143, 259 and 302 workers (Fig 1). The three companies belonged to commercial activities and services sector.

Table 1 shows workers' stress and job satisfaction over time by job position with the difference between the two populations, and the distribution of missing values. Additional information on the distribution of variables can be found in Multimedia Appendix 1 (Fig S1 in S1 Appendix, Table S1 in S1 Appendix).

### Influence of job position on average stress and job satisfaction

Univariate analyses revealed that stress was not different between job position (g = 0.13, 95CI −0.10 to 0.36, p = 0.3), but that job satisfaction was lower for employees than for managers (−0.65, −0.89 to −0.42, p<0.001) (Fig 2). In managers, neither age, seniority nor gender influenced the two indicators. Employees with more than five years of seniority had higher stress (0.26, 0.10 to 0.43, p = 0.004) and lower job satisfaction (−0.33, −0.50 to −0.16, p = 0.002), while age and gender did not show any influence on either indicator. The sensitivity analysis confirmed these results (S3 Table 1 in S3 Appendix, S3 Table 2 in S3 Appendix).

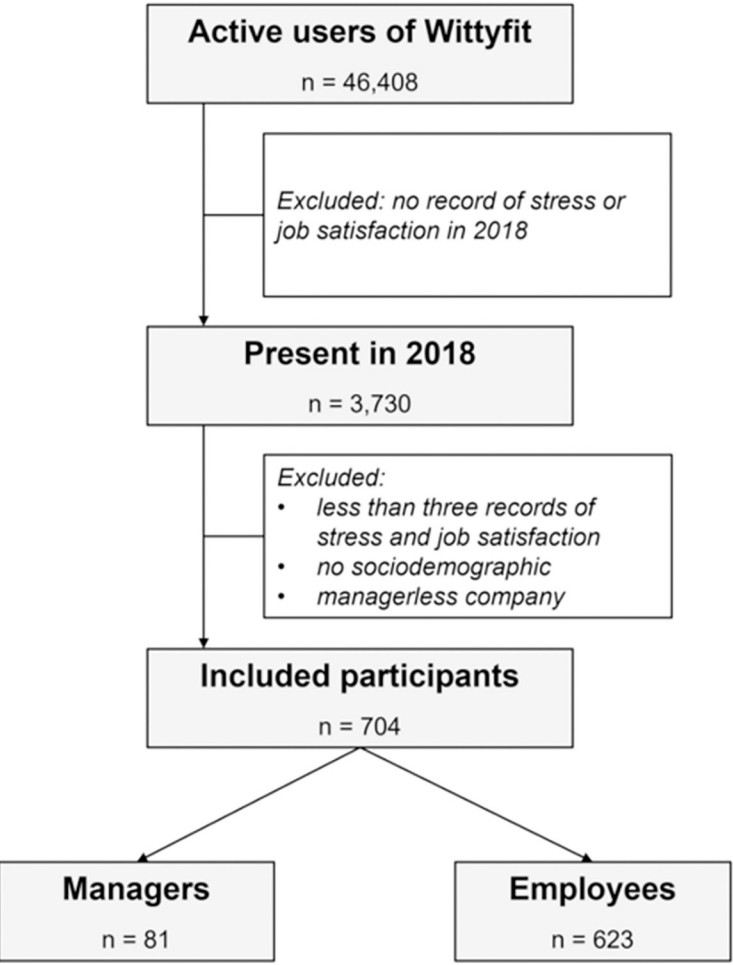

**Fig 1. Flowchart of the study.** Legend: 'n': sample size.

**Table 1. Stress and job satisfaction by job position over time.**

| Outcome (by time) | Manager Mean±SD (NA/n) | Employee Mean±SD (NA/n) | Difference between groups (g and 95CI) |
|---|---|---|---|
| Stress (/100) | | | |
| 2018 | 53.7±26.3 (1/81) | 51.6±27.7 (8/623) | −0.05 (−0.61 to 0.32) |
| 2019 | 49.3±23.5 (17/81) | 52.8±27.8 (140/623) | 0.06 (−0.44 to 0.83) |
| 2020 | 45.9±29.0 (24/81) | 50.5±25.2 (157/623) | 0.13 (−0.19 to 1.09) |
| 2021 | 44.2±27.4 (20/81) | 50.7±26.7 (149/623) | 0.17 (−0.04 to 1.24) |
| Job satisfaction (/100) | | | |
| 2018 | 69.3±22.0 (0/81) | 57.5±25.2 (1/623) | −0.28 (−1.26 to −0.46) |
| 2019 | 70.4±19.5 (16/81) | 58.3±26.7 (122/623) | −0.32 (−1.55 to −0.46) |
| 2020 | 72.0±24.4 (24/81) | 61.5±25.2 (158/623) | −0.29 (−1.50 to −0.32) |
| 2021 | 76.6±18.5 (19/81) | 61.2±26.3 (148/623) | −0.42 (−1.87 to −0.82) |

Legend: Difference in stress and job satisfaction are calculated between managers and employees at each time using univariate linear random-effect model with "company" effect as random. A bold result means that the difference is significant in terms of effect size. 'SD': standard deviation, 'NA': missing data (not available), 'n': group size (number), 'g': Hedges' g effect size, '95CI': 95% confidence interval.

**Univariate and bivariate random-effect models on cross-sectional data**
Differences in stress and job satisfaction by job position

Legend: '***': *P*<.001, 'NS': not significant.

The bivariate model revealed that managerial position can influence workers' job satisfaction, but not their stress. Effect size measuring the difference in job satisfaction by job position, calculated using Hedge's g, was found to be moderate (Hedges' g=−0.65, 95CI −0.89 to −0.42, *P*<.001). No differences in mean stress level were found between groups (0.13, −0.10 to 0.36, *P*=.3).

**Fig 2. Stress and job satisfaction by job position across all years (cross-sectional univariate and bivariate analyses).**

Overall, bivariate cross-sectional analysis revealed that the job satisfaction was lower in employees (−0.85, −1.10 to −0.62, p<0.001), and stress not different (0.14, −0.11 to 0.36, p = 0.2). Age was found to reduce job satisfaction of managers (−0.44, −0.89 to −0.01, p = 0.05), but not seniority or gender. In employees, after cross-sectional analysis, workers with greater than five-years seniority were found to be more stressed (0.34, 0.18 to 0.52, p<0.001) and less satisfied (−0.47, −0.63 to −0.30, p<0.001). Age and gender had no effect on indicators. The results of the sensitivity analyses, except for some obtained with the complete cases analysis were in line with these results (S3 Table 3 in S3 Appendix).

## Average changes in stress and job position over time

Over time, while workers' stress has not changed (p = 0.3), their job satisfaction did. After one year of stability, it improved slightly in 2020 (g = 0.13, 95CI 0.02 to 0.24, p = 0.02) and 2021 (0.17, 0.07 to 0.28, p = 0.002) (Fig 3A). In managers, while there was no change in 2019 and 2020, stress was found to be lower in 2021 (−0.31, −0.63 to −0.01, p = 0.05), and job satisfaction higher (0.34, 0.05 to 0.66, p = 0.03). Only the results obtained with the multiple-imputed data-set can confirm these results. For the entire sample, employees' job satisfaction improved very slightly in 2020 (0.14, 0.04 to 0.27, p = 0.006) and 2021 (0.16, 0.05 to 0.29, p<0.001) (Fig 3B).

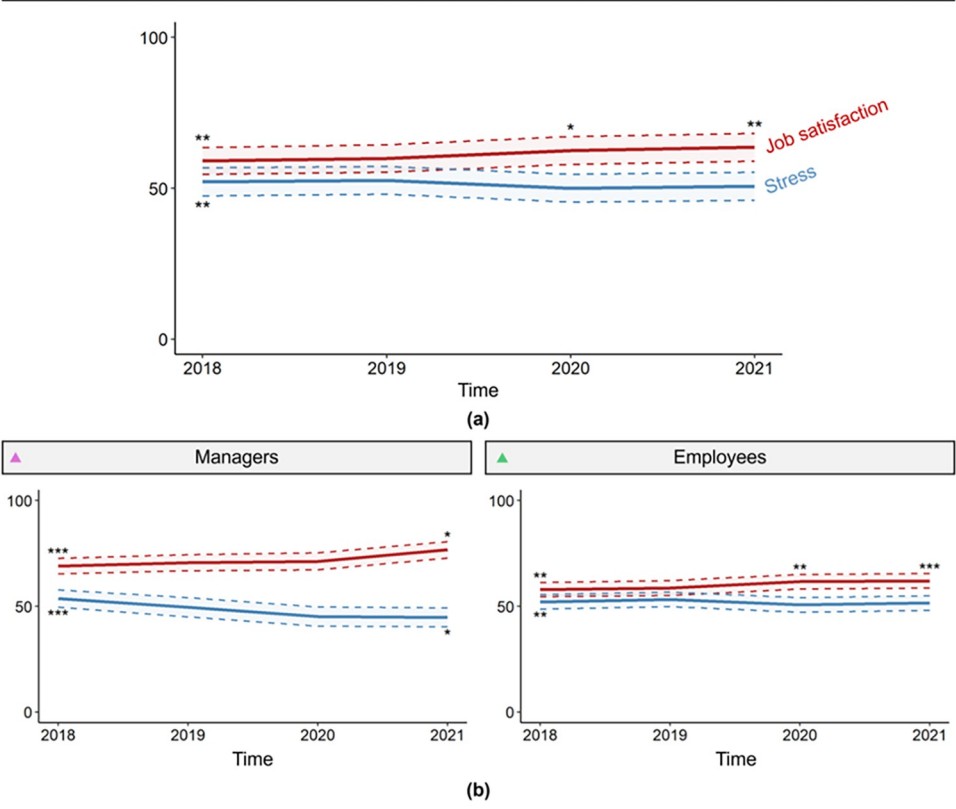

**Fig 3. Evolution of stress and job satisfaction over time by job position (longitudinal bivariate analysis).** Legend: (a) for all workers, (b) by job position.

Furthermore, the comparison of average changes by job position revealed that stress was not different in any year (p = 0.3), but that job satisfaction for employees remained lower than for managers over time (p<0.001). Indeed, a small difference was found in 2020 (–0.47, –0.79 to –0.14, p = 0.002), and a moderate difference in 2018 (–0.68, –0.99 to –0.38, p<0.001), 2019 (–0.63, –0.95 to –0.32, p<0.001) and 2021 (–0.74, –1.05 to –0.41, p<0.001). The results of the sensitivity analyses, except for some obtained with the complete cases analysis, were in line with these results (S3 Table 4 in S3 Appendix).

Managers over 40 reported being more stressed (g = 0.45, 95CI 0.02 to 0.88, p = 0.04) and less satisfied (–0.44, –0.90 to –0.02, p = 0.04) with their job. Employees with more than five years' seniority were also found to be more stressed (0.36, 0.20 to 0.52, p<0.001) and less satisfied (–0.46, –0.62 to –0.28, p<0.001). Overall, the results of the sensitivity analyses, except for some obtained with the complete cases analysis, confirmed these results (S3 Table 5 in S3 Appendix).

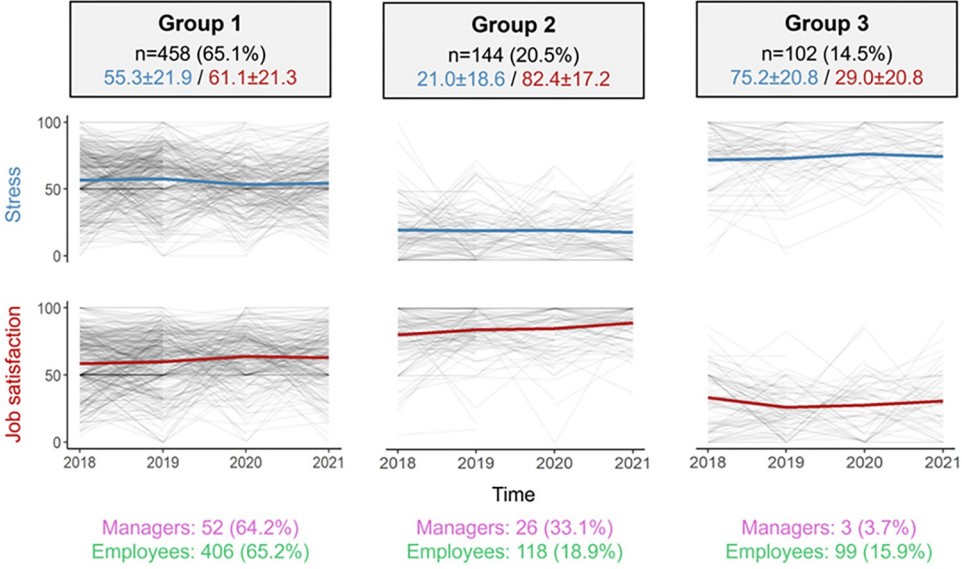

**Fig 4. Group-based multi-trajectories of workers' stress and job satisfaction.**

### Individual multi-trajectory of stress and job satisfaction among workers

Three latent groups of workers were found in the global sample (entropy = 0.80) (Fig 4). The results of the sensitivity analysis, except for the complete-cases analysis, confirmed this optimal number of latent groups (S3 Table 6 in S3 Appendix). Group 1 (n = 458 individuals, 65.1% of the sample) included workers whose stress (55.3±21.9) and job satisfaction (61.1±21.3) were moderate over time, on average, with a slight reduction in stress in 2020 (g = −0.19, 95CI−0.33 to −0.05, p = 0.008), and an increase in job satisfaction in 2020 (0.26, 0.11 to 0.39, p<0.001) and 2021 (0.25, 0.11 to 0.40, p = 0.002) compared to 2018. Group 2 (n = 144, 20.5%) included workers whose stress remained low and stable over time (21.0±18.6) and job satisfaction was high (82.4±17.2), on average, with an increase in 2021 (0.40, 0.13 to 0.63, p = 0.004). Group 3 (n = 102, 14.5%), which included workers whose stress was high over time (75.2±20.8) while job satisfaction was low (29.0±20.8), on average, with a decrease in 2019 (−0.45, −0.75 to −0.16, p<0.001) and 2020 (−0.29, −0.58 to 0.00, p = 0.05).

The proportion of employees and managers in each latent group was found to be different (p<0.001). Indeed, although it seemed equivalent in group 1 (managers 64.2% vs. employees 65.2%), the proportion of managers appeared to be higher than that of employees in group 2 (33.1% vs. 18.9%), and lower in group 3 (3.7% vs. 15.9%). The results of sensitivity analysis confirmed these results (S3 Table 6 in S3 Appendix).

### Cross-lagged effects of stress and job satisfaction by job position

Overall, stress and job satisfaction of workers influenced each other over time (p<0.001). The stress of workers in 2018 seemed to have slightly reduced their job satisfaction in 2019 (g = −

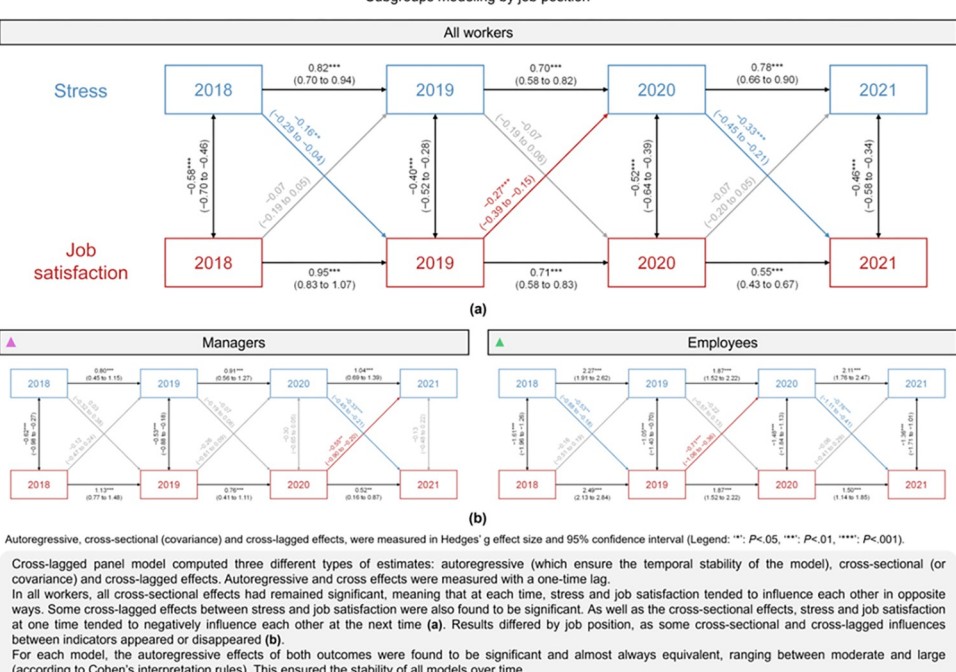

**Fig 5. Cross-lagged effects of stress and job satisfaction by job position.** Legend: (a) for all workers, (b) by job position.

0.16, 95CI –0.29 to –0.04, p = 0.006). This was also found from 2020 to 2021 (–0.33, –0.45 to –0.21, p<0.001). Simultaneously, workers' job satisfaction did slightly reduce their stress from 2019 to 2020 (–0.27, –0.39 to –0.15, p<0.001). In addition to cross-lagged effects, all cross-sectional influences between stress and job satisfaction were found to be significant and negative (p<0.001) (Fig 5A). The sensitivity analysis confirmed these results (S3 Table 7 in S3 Appendix).

When analyzing by job position, similar results were found in employees, but not in managers. Indeed, in managers, only the last two cross-lagged effects were significant, namely that of stress and job satisfaction (–-0.33, –0.45 to –0.21, p<0.001), and conversely (–0.55, –0.90 to –0.20, p = 0.002). Moreover, only the first two covariances between stress and job satisfaction appeared to be significant (2018: –0.62, –0.98 to –0.27, p<0.001; 2019: –0.53, –0.88 to –0.18, p = 0.003) (Fig 5B). The results of the sensitivity analyses confirmed these results (S3 Table 7 in S3 Appendix).

## Discussion

The main results showed that the managerial position was mainly a factor in improving job satisfaction among workers, with almost no effect on stress. Socio-demographic factors such as age and seniority may have influenced these two indicators, but not gender. Overall, workers' job satisfaction improved and in parallel, their stress did not change, even though the two indicators had a negative influence on each other over time, however, these results were different according to job position.

### Stress, job satisfaction and the managerial role

Our findings suggest that the managerial position is mainly a factor in improving job satisfaction, but not stress. This suggests that the responsibilities associated with the position do not

add stress for managers. A study among Norwegian employees found that the job strain risk, which can cause stress to workers [32,33], was not different between regular employees and workers with managerial roles [34]. Despite higher demands, greater control protects managers from this risk [18,35]. A study in nurses showed that managers may face higher occupational stressors [36]. Like previous research, the role within the working environment makes managers more satisfied with their work than employees. A study in the emergency department health professionals highlighted that job position played a role in workers' holistic satisfaction [16]. Another study in nursing personnel also found that job satisfaction may differ by job position, with nurse managers having the highest job satisfaction of all groups [37]. Organizations should strive to enhance the job satisfaction of their workers, especially employees, as a factor in job well-being [12,38–40].

## Mean changes and group-based multi-trajectories of stress and job satisfaction over time

On average, while their stress remained stable, the job satisfaction of workers was above the baseline in the last two years of the study. While employees followed the same trend, we found that managers experienced different changes for both outcomes. The last year, at the same time as their job satisfaction improved compared to baseline, their stress also decreased. Three latent groups with specific evolutionary trajectories of stress and job satisfaction were identified in the sample. Groups can be named according to the shape of their associated multi-trajectory as follows: the average (group 1), the good (group 2) and the poor (group 3) occupational health groups. The proportion of managers and employees was different across groups, with notably a higher proportion within the good occupational health group. These two results suggest that the managerial position may act as a long-term factor in reducing stress and improving job satisfaction. Further analyses of the average changes in managers and the individual multi-trajectories of group 2 and group 3 especially, including the use of static covariates, could reveal the protective and risk factors associated with good occupational health of workers [41].

## Concurrent influences between stress and job satisfaction

The negative reciprocal influences between stress and job satisfaction have been studied and identified many times, both transversally [11,12,14] and longitudinally [13]. In our study, in addition to cross-sectional reciprocal influences, cross-lagged influences were also highlighted. This suggests that stress and job satisfaction can influence each indicator over time. For example, a satisfied worker might experience a reduction in stress because of the benefits of good job satisfaction [6]. In contrast, a stressed worker may experience a decrease in job satisfaction [10,11]. Among managers, we found that indicators were interconnected in the last two years. This could explain the changes observed on average in this population, where stress and job satisfaction have evolved in opposite directions over the final year. These results are similar to a previous study in nurses [42]. This suggests that to improve occupational health through stress and job satisfaction, organizations should strive to implement holistic interventions to make them more effective.

## Sociodemographic factors of occupational health among managers and employees

Sociodemographics factors were found to influence managers' and employee's indicators. The results of the longitudinal analysis suggest that managers over 40 years old had poorer

occupational health, reporting higher stress and lower job satisfaction than others. This finding is consistent with literature that the risk of stress is greater among older workers [43], and that age and job satisfaction are negatively related [44]. The same result was found for employees with more than five years of seniority. Regardless of the job position, no influence of gender was found, suggesting that male and female workers face the same putative occupational health risk factors. Organizations must therefore consider the personal characteristics of workers to ensure the effectiveness of interventions and programs related to improving workplace health.

## Limitations

We acknowledge some limitations in our study. Workers' self-reported responses to questionnaires may have resulted in self-report bias [45]. This could also have led to an affective bias caused by social desirability [46]. However, Wittyfit allows its users to express themselves freely by anonymizing all user data, thus reducing this bias. The process of averaging the records may have hidden a potential seasonal bias. Nevertheless, the distribution of records, reported in S4 Table 1 in S4 Appendix, appeared very arbitrary, revealing no seasonal effect. The COVID–19 pandemic in France may have affected the responses given by workers in 2020 (or even 2021). On the one hand, we found, using the linear mixed model with time effect treated as random, that the differences in stress and job satisfaction between managers and employees remained consistent before, during, and after 2020. On the other hand, the temporal stability provided by the cross-sectional panel analysis tends to prove that the influence of COVID–19 pandemic was negligible. Although stress and job satisfaction are known to be associated with well-being [47] and may therefore appear as an important covariate, we did not measure it. The study suffered from a lack of information on workers' socio-demographic characteristics, such as employment, level of education or income. It cannot be ruled out that workers with a low level of education or limited computer skills may have been excluded. As it is well documented that the level of job stress and job satisfaction varies by the work sector for both employees and managers [48], our results could not be generalized to whole working population. The low sample size also precluded generalizability. However, the database grows over the years [22], and further studies with larger populations and longer time periods could be designed to confirm or add to these initial results. It would also be very valuable to compare the results of this study, based on a real-world database and therefore constitutes real evidence, with those of a randomized clinical trial [41].

## Conclusions

Although they do not perceive a difference in stress, managers have higher job satisfaction than employees. The changes in both indicators over time, about job position and other socio-demographic factors, may have long-term effects on occupational health of workers. Moreover, the latent evolutionary trajectories followed by some of the workers suggest that by understanding their putative drivers, organizations may be able to improve the occupational health of all their workers in a holistic way.

## Supporting information

**S1 Appendix. Distribution of outcomes.**
(DOCX)

**S2 Appendix. Statistical analysis plan.**
(DOCX)

**S3 Appendix. Sensitivity analysis results.**
(DOCX)

**S4 Appendix. Reporting guidelines.**
(DOCX)

**S1 Data.**
(ZIP)

# Acknowledgments

We express our sincere gratitude to all voluntary workers using Wittyfit, who participated in this study.

# Author Contributions

**Conceptualization:** Rémi Colin-Chevalier, Frédéric Dutheil, Samuel Dewavrin, Thomas Cornet.

**Data curation:** Samuel Dewavrin, Thomas Cornet.

**Formal analysis:** Rémi Colin-Chevalier.

**Investigation:** Rémi Colin-Chevalier.

**Methodology:** Rémi Colin-Chevalier, Frédéric Dutheil, Bruno Pereira.

**Project administration:** Rémi Colin-Chevalier.

**Resources:** Rémi Colin-Chevalier.

**Software:** Rémi Colin-Chevalier.

**Supervision:** Frédéric Dutheil, Bruno Pereira.

**Validation:** Rémi Colin-Chevalier, Frédéric Dutheil, Amanda Clare Benson, Samuel Dewavrin, Thomas Cornet, Céline Lambert, Bruno Pereira.

**Visualization:** Rémi Colin-Chevalier.

**Writing – original draft:** Rémi Colin-Chevalier.

**Writing – review & editing:** Rémi Colin-Chevalier, Frédéric Dutheil, Amanda Clare Benson, Samuel Dewavrin, Thomas Cornet, Céline Lambert, Bruno Pereira.

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
