## [Decision Letter · Decision Letter 0]

6 Sep 2023

PONE-D-23-14513Stress and job satisfaction over time, the influence of the managerial position: a bivariate longitudinal modelling of Wittyfit dataPLOS ONE

Dear Dr. Colin-Chevalier,

Thank you for submitting your manuscript to PLOS ONE. After careful consideration, we feel that it has merit but does not fully meet PLOS ONE’s publication criteria as it currently stands. Therefore, we invite you to submit a revised version of the manuscript that addresses the points raised during the review process.

We look forward to receiving your revised manuscript.

Kind regards,

Prakash K.C., Ph.D.

Academic Editor

PLOS ONE

Journal Requirements:

"I have read the journal's policy and the authors of this manuscript have the following competing interests:

RC, SD, and TC are part of Wittyfit. Other authors have declared that no competing interests exist (FD is responsible for the scientific accuracy of Wittyfit but is not paid by Cegid; as previously published, Wittyfit is a public private partnership with the CHU of Clermont-Ferrand, France)."

Please confirm that this does not alter your adherence to all PLOS ONE policies on sharing data and materials, by including the following statement: ""This does not alter our adherence to  PLOS ONE policies on sharing data and materials.” (as detailed online in our guide for authors http://journals.plos.org/plosone/s/competing-interests).  

If there are restrictions on sharing of data and/or materials, please state these. Please note that we cannot proceed with consideration of your article until this information has been declared. 

4. We notice that your supplementary files [Appendix S1 to S4] are included in the manuscript file. Please remove them and upload them with the file type 'Supporting Information'. Please ensure that each Supporting Information file has a legend listed in the manuscript after the references list.

**Additional Editor Comments:**

Psychological wellbeing at work is an important aspect of working life connected to overall wellbeing and health of an individual. Being topical it has gained a lot of research attention in recent decades.

Work-related psychosocial demands are highlighted as key aspects to employee wellbeing in existing literatures. This is a well written manuscript, which could be a good addition to the current body of scientific literatures and comes up with important

dimension that is rarely examined and I am impressed by the analyses. However, there are few important aspects missing in the manuscript, which are nicely summarized by Reviewer 1. Given the importance of this research, I would like to

give the authors a chance of revision. I strongly recommend major revisions based on the comments given by reviewer 1.

Reviewers' comments:

Reviewer's Responses to Questions

**Comments to the Author**

1. Is the manuscript technically sound, and do the data support the conclusions?

Reviewer #1: Partly

Reviewer #2: Yes

2. Has the statistical analysis been performed appropriately and rigorously? 

Reviewer #1: Yes

Reviewer #2: Yes

3. Have the authors made all data underlying the findings in their manuscript fully available?

Reviewer #1: Yes

Reviewer #2: Yes

4. Is the manuscript presented in an intelligible fashion and written in standard English?

Reviewer #1: Yes

Reviewer #2: Yes

5. Review Comments to the Author

Reviewer #1: In this article, the authors have explored the bivariate effect of managerial position on stress and job satisfaction of the employees and the inter-relationship of these indicators over time. The influence of age, seniority, and gender on stress and job satisfaction was also assessed. This is an important topic as stress is becoming increasingly common in the everyday life of the modern workforce and might have a negative impact on job satisfaction. Increased job stress and low job satisfaction not only impact the individual’s overall well-being but also impacts the work organization and families. The scientific writing is good, the statistical analysis is nicely done, and the results are presented well. However, the are some major concerns:

The sample size presented in the study is quite small and belongs to three companies. How are the companies selected? What sector of work do these companies belong to? It is well documented that the level of job stress and job satisfaction varies by the work sector for both employees and managers. Hence, it could not be generalized to whole working population without having sufficient background information on the work sector.

What kind of questionnaires were included in the measurement of job satisfaction and job stress? Or is it just the two components "stress" and "job satisfaction" and a visual analog scale from 0-100 to report the perceived levels? The job stress for example is a broad concept and determined based on several components such as job position, age, seniority, and gender, for example, workplace conditions, kind of work (regular or irregular), workload, etc.

The study does not include the basic characteristics of participants such as socio-economic conditions such as income and education which is an important factor for job stress and job satisfaction. Moreover, without information on educational background and the sector of work, it could be expected that the employees with lower levels of education and the ones who are not comfortable using the software to answer the questions would systematically be excluded from the study.

The reduced level of job stress is associated with the perceived psychological well-being/ general well-being of a person and ultimately impacts the level of job satisfaction. There is not any measurement of the general perceived well-being of the participants included in the study. Since the Wittyfit software is designed to evaluate overall health, the health and well-being-related information would be an important covariate/ indicator in this study.

I would recommend addressing these major issues in the manuscript in order for the manuscript for it to have a better scientific impact and suitable for publication in the journal.

Reviewer #2: The manuscript is well written. In the discussion component while highlighting the limitations of the study, please kindly justify how this original research adds to existing knowledge base considering the already existing evidence in this area of research. the limitations of the study need to be clearly highlighted with few justifications. The recent published literature which can be considered for a comparative analysis (although it is not exactly what you covered in the research but has overall general theme of relevance) include:

Dutheil F, Duclos M, Naughton G, Dewavrin S, Cornet T, Huguet P, Chatard JC, Pereira B. WittyFit—Live Your Work Differently: Study Protocol for a Workplace-Delivered Health Promotion. JMIR Research Protocols. 2017 Apr 13;6(4):e6267.

Colin-Chevalier R, Dutheil F, Cambier S, Dewavrin S, Cornet T, Baker JS, Pereira B. Methodological issues in analyzing real-world longitudinal occupational health data: a useful guide to approaching the topic. International Journal of Environmental Research and Public Health. 2022 Jun 8;19(12):7023.

Vialatte L, Pereira B, Guillin A, Miallaret S, Baker JS, Colin-Chevalier R, Yao-Lafourcade AF, Azzaoui N, Clinchamps M, Bouillon-Minois JB, Dutheil F. Mathematical modeling of the evolution of Absenteeism in a University hospital over 12 years. International Journal of Environmental Research and Public Health. 2022 Jul 6;19(14):8236.

6. PLOS authors have the option to publish the peer review history of their article (what does this mean?). If published, this will include your full peer review and any attached files.

Reviewer #1: No

Reviewer #2: No

---

## [Author Response · Author response to Decision Letter 0]

28 Dec 2023

All comments from reviewers and editors have been addressed and are the subject of a letter entitled "Response to Reviewers" attached.

---

## [Editor Report · Decision Letter 1]

21 Jan 2024

Stress and job satisfaction over time, the influence of the managerial position: a bivariate longitudinal modelling of Wittyfit data

PONE-D-23-14513R1

Dear Dr. Colin-Chevalier,

We’re pleased to inform you that your manuscript has been judged scientifically suitable for publication and will be formally accepted for publication once it meets all outstanding technical requirements.

Kind regards,

Prakash K.C., Ph.D.

Academic Editor

PLOS ONE
---

## [Editor Report · Acceptance letter]

23 Feb 2024

PONE-D-23-14513R1 

PLOS ONE

Dear Dr. Colin-Chevalier, 

I'm pleased to inform you that your manuscript has been deemed suitable for publication in PLOS ONE. Congratulations! Your manuscript is now being handed over to our production team.

Kind regards, 

on behalf of

Dr. Prakash K.C. 

Academic Editor

PLOS ONE